# Role of Extracellular Vesicle-Based Cell-to-Cell Communication in Multiple Myeloma Progression

**DOI:** 10.3390/cells10113185

**Published:** 2021-11-16

**Authors:** Ilaria Saltarella, Aurelia Lamanuzzi, Benedetta Apollonio, Vanessa Desantis, Giulia Bartoli, Angelo Vacca, Maria Antonia Frassanito

**Affiliations:** 1Department of Biomedical Sciences and Human Oncology, Unit of Internal Medicine “Guido Baccelli”, University of Bari Medical School, Piazza Giulio Cesare 11, 70124 Bari, Italy; ilaria.saltarella@uniba.it (I.S.); aurelia.lamanuzzi@uniba.it (A.L.); benedettaapollonio@gmail.com (B.A.); vanessa.desantis@uniba.it (V.D.); giulia23bartoli@gmail.com (G.B.); angelo.vacca@uniba.it (A.V.); 2Department of Biomedical Sciences and Human Oncology, Pharmacology Section, University of Bari Medical School, Piazza Giulio Cesare 11, 70124 Bari, Italy; 3Department of Biomedical Sciences and Human Oncology, Unit of General Pathology, University of Bari “Aldo Moro”, 70124 Bari, Italy

**Keywords:** multiple myeloma, extracellular vesicles, bone marrow microenvironment, miRNAs

## Abstract

Multiple myeloma (MM) progression closely depends on the bidirectional crosstalk between tumor cells and the surrounding microenvironment, which leads to the creation of a tumor supportive niche. Extracellular vesicles (EVs) have emerged as key players in the pathological interplay between the malignant clone and near/distal bone marrow (BM) cells through their biologically active cargo. Here, we describe the role of EVs derived from MM and BM cells in reprogramming the tumor microenvironment and in fostering bone disease, angiogenesis, immunosuppression, drug resistance, and, ultimately, tumor progression. We also examine the emerging role of EVs as new therapeutic agents for the treatment of MM, and their potential use as clinical biomarkers for early diagnosis, disease classification, and therapy monitoring.

## 1. Introduction

Multiple myeloma (MM) is a hematologic neoplasm characterized by the expansion of monoclonal malignant plasma cells (MM cells) that accumulate in the bone marrow (BM). MM usually progresses from the preneoplastic and asymptomatic phases of monoclonal gammopathy of undetermined significance (MGUS) and smoldering myeloma (SMM) to symptomatic MM [1]. Despite the development of new drugs, MM is still an incurable malignancy. Several studies have shown that the crosstalk between cancer cells and BM microenvironment plays a crucial role in disease progression and drug resistance, with the tumor being able to hijack surrounding cells through various mechanisms of cell-to-cell communication, including autocrine/paracrine signaling and the release of extracellular vesicles (EVs) [2].

Based on their size, EVs were first classified into apoptotic/necrotic bodies (size 1000–5000 nm), microvesicles (200–1000 nm) and exosomes (30–150 nm) [3]. Regarding apoptotic/necrotic bodies originate from apoptotic and necrotic cells that break up into multiple vesicles, microvesicles are pinched off from the plasma membrane, while exosomes are formed by an active, energy-dependent and organized process [4]. Currently, the MISEV18 guidelines recommend the use of the term “EVs” for a heterogeneous family of vesicles (both small and medium/large vesicles) delimited by a lipid bilayer that are released from all eukaryotic cells in various body fluids (plasma, urine, saliva, and cerebrospinal fluid) [5].

Although exosomes were initially regarded as cellular waste with no biological activities [6], it is now widely accepted that EVs play an essential role in intercellular communication via their cargo, capable of reprogramming near and distal recipient cells [3,7].

EVs share multiple molecules with the parental cell, providing useful information about the genetic, molecular, and functional properties of the cell of origin [8,9]. Tumor cells produce and release a greater amount of EVs (tumor-derived EVs) than healthy proliferating cells [8]. Although some proteins involved in EVs biogenesis (i.e., p53 [10] and Rab proteins [11]) can be altered during tumor progression, the mechanisms leading to increased production and secretion of EVs by tumor cells are still unclear.

In recent years, several studies have documented the central role of EVs as key regulators of the bidirectional crosstalk between tumor cells and BM microenvironment during disease progression.

Here we provide an overview of the role of EVs derived from MM and BM cells in disease progression, drug resistance, bone resorption, immunosuppression and angiogenesis. We also discuss the emerging role of EVs as new therapeutic agents for the treatment of MM, and their potential use as clinical biomarkers for early diagnosis, disease classification, and therapy monitoring.

## 2. EVs Biogenesis

Two main pathways of biogenesis have been described for the generation of EVs: (i) direct budding of the plasma membrane with consequent release of EVs, classically referred to as microvesicles [4,12]; and (ii) endosomal biogenesis, an active and complex mechanism that ultimately leads to the production of EVs, classically referred to as exosomes (Figure 1A) [13].

Endosomal biogenesis begins with the inward budding of plasma membrane, giving rise to an early endosome, which in turn undergoes multiple invaginations leading to the formation of intraluminal vesicles (ILVs) incorporated into a mature multivesicular body (MVB) (Figure 1A). As a result, the ILVs express on their surface various proteins belonging to the plasma membrane of the cell of origin, and they enclose the parental cytosol.

EVs contain endosome-related membrane transport and fusion proteins that contribute to EVs biogenesis (annexins, flotillin, GTPases, and lipids, such as ceramides and sphingomyelin) [14,15], tetraspanins (CD9, CD63, CD81, and CD82) used as EVs markers [5,16], and proteins belonging to the accessory endosomal sorting complex required for transport (ESCRT) [17,18]. Activation of ESCRT pathway is responsible for EVs sorting, packaging, and transport [18]. Both the ESCRT machinery and its auxiliary proteins (Alix, Vps4, and VTA1) are localized on the cytoplasmic side of the endosomal membrane and are involved in the sorting and ubiquitination of proteins into ILVs. In particular, ESCRT-0 recognizes and binds to ubiquitinated proteins, while ESCRT-I interacts with ESCRT-II, forming another complex. The two complexes bind to ESCRT-III, which is involved in the budding process, and induces the detachment of ILVs from the MVB membrane by activating the ATPase protein Vps4 [19,20]. Alix associated with ESCRT proteins is involved in endosomal membrane budding and abscission, and cargo selection via syndecan binding [21,22].

In addition, two ESCRT-independent pathways have been identified: (i) tetraspanin-enriched domains (TEMs)-dependent, where tetraspanins act as sorting machinery [23], and (ii) sphingolipids-dependent, where ceramides trigger spontaneous negative curvature of the endosomal membrane [24].

At the end of the maturation process, the newly formed MVBs can fuse either with lysosomes for degradation or with plasma membranes for the release of EVs into the extracellular space [25] (Figure 1A). The release process is regulated by transmembrane proteins (e.g., Rab, N-ethylmaleimide-sensitive factor-NSF, and soluble NSF-attachment protein-SNAP) and membrane complexes (e.g., SNAP-attachment protein receptor-SNARE) [11,26,27]. SNARE proteins, localized on different intracellular membranes, are usually responsible for the fusion of different cellular compartments [28], and trigger the release of EVs through the formation of invadopodia [12,29].

When released into the extracellular space, EVs interact with local cells or enter the blood and lymphatic circulation, allowing interaction with distant cells [30].

Several mechanisms regulate the interaction of EVs with target cells, including fusion, internalization, micropinocytosis [31], and phagocytosis [32]. Interestingly, fusion of EVs is regulated by the same proteins involved in their release, i.e., SNAREs and Rab proteins, while internalization of EVs requires clathrin- and caveolin-mediated endocytosis [33,34], integrins [35], and membrane-associated lipids, including cholesterol, sphingolipids, and glycosylphosphatidylinositol (GPI)-anchored proteins [36]. Moreover, surface antigens of EVs (i.e., MHC-I/-II, tetraspanins, Fas ligand (FasL), and TNF related apoptosis inducing ligand (TRAIL)) can interact with their specific receptors on target cells and activate intracellular signaling [4,37,38] (Figure 1B).

## 3. EVs Cargo

The biological activity of EVs is closely dependent on their cargo, such as proteins, lipids, and nucleic acids, which are responsible for target cell reprogramming and provide important information about the parental cells by mirroring their cytoplasmic content (Figure 1B).

EVs proteins include endosomal, cytosolic, and nuclear proteins [39], involved in EVs biogenesis, transport, and fusion (e.g., HSP70, HSP90), integrins and adhesion molecules that play a role in target cells binding [40,41,42]. In addition, under pathological conditions, other molecules may also be included in the cargo of EVs. For example, tumor-derived EVs contain specific oncoproteins (HER family [43]) and immunosuppressive molecules (Fas-L, TRAIL, and immune checkpoints such as PD-L1 [44,45]) that promote neoplastic progression and immune evasion (Figure 1B).

Recent studies have shown that EVs can be ‘decorated’ with additional proteins besides the canonical exofacial molecules, collectively referred to as ‘corona’ [46,47,48,49,50,51,52]. Buzas et al. [46] showed that EVs associate with extracellular matrix proteins, complement, immunoglobulins, coagulation factors, lipoproteins, nucleic acids, and thiol-reactive antioxidants [46]. Furthermore, Toth et al. [51] confirmed the interaction between plasma proteins and EVs, and identified a number of proteins (ApoA1, ApoB, ApoC3, ApoE, complement factors 3 and 4B, fibrinogen α-chain, immunoglobulin heavy constant γ2 and γ4 chains) that form a ‘corona’ around EVs in blood plasma [51].

EVs also carry nucleic acids as DNAs (single-stranded, double-stranded, genomic, mitochondrial and reverse-transcribed complementary DNA) [53,54,55,56] and RNAs, including mRNAs and non-coding RNAs (microRNAs (miRs), long non-coding RNAs (lncRNAs), circular RNAs (circRNAs), tRNA-derived small RNA fragments, and YRNAs; for recent and detailed review see [57]) (Figure 1B). The lipid bilayer of EVs protects RNA from degradation and increases its stability. Several studies have highlighted the important role of the RNA content of EVs in modulating the transcriptome of target cells and in shaping the tumor microenvironment [58,59,60,61].

Since the molecular and genetic cargo of EVs partially reflects the composition of parental cells and EVs can be easily obtained from patients’ serum/plasma or other body fluids, several studies suggest that tumor-derived EVs may be useful for cancer diagnosis, prognosis and drug responses (see ‘Diagnostic potential’ section) [52,62,63,64].

## 4. EVs in MM Progression and Drug Resistance

MM depends on the BM milieu that co-evolves with the tumor and promotes cancer cell proliferation, drug resistance, and disease progression by inducing bone resorption, immunosuppression, and angiogenesis [65] (Figure 2). Among BM stromal cells (BMSCs), fibroblasts (FBs) contribute to the formation of the tumor supportive niche and promote disease progression and resistance to bortezomib-induced apoptosis [66,67]. Frassanito et al. [68] have shown that EVs promote the bidirectional crosstalk between MM cells and FBs in the tumor microenvironment [68].

Specifically, MM-derived EVs (MM-EVs) contain the WWC2 protein, which activates the Hippo pathway, which in turn triggers a de novo synthesis of miR-27b-3p and miR-214-3p into recipient FBs [68]. Moreover, MM-EVs activate FBs by increasing the expression of fibroblast activation protein (FAP) and alpha-smooth muscle actin (αSMA) promoting their transformation into cancer associated fibroblasts (CAFs). Taken together, these data support evidence that MM-EVs can directly reprogram surrounding cells in the tumor microenvironment during disease progression. Similarly, Cheng et al. [69] demonstrated that EVs derived from the OPM2 MM cell line contain high levels of miR-21 and miR-146, which trigger cell proliferation, IL-6 release, and conversion of mesenchymal stromal cells (MSCs) into CAFs, as documented by overexpression of FAP, αSMA, and SDF-1 [69]. Knockdown of both miRs in MSCs prevents their transformation and IL-6 release. Moreover, MM-EVs induce secretion of IL-6 by BMSCs via activation of APE1/NF-kB pathway [70]. Finally, EVs from both plasma and MM cells of patients promote cell proliferation, migration, and adhesion of recipient BMSCs by transferring multiple oncogenic factors highlighting the role of MM-EVs in modifying the BM niche [71].

On the other hand, MSCs-EVs from MM patients and healthy donors (HD) differentially affect the phenotype and the intracellular signaling of MM cells: MM MSCs-EVs enhance viability, proliferation, and migration of MM cells, while HD MSCs-EVs do not [72].

EVs from BMSCs support myeloma progression by delivering miRs, pro-survival cytokines and chemokines [73]. Roccaro et al. [74] demonstrated that MSCs-EVs contain oncogenic proteins, cytokines and adhesion molecules, including IL-6, MCP-1, junction plakoglobin, fibronectin, and express low level of the tumor-suppressive miR-15a compared to EVs derived from healthy MSCs, which promote tumor growth and dissemination of MM cells in vivo [74]. Recently, proteomic analysis of MSCs-EV cargo revealed that MM MSCs-EVs heterogeneously express higher levels of ribosomal proteins than their HD counterpart. EVs enriched with ribosomal proteins promote the viability and proliferation of MM cells [72,75]. Moreover, EVs derived from the BM of 5T33 mice contain pro-tumor proteins, including MCP-1, MIP-1α, and SDF-1, which promote proliferation, survival, and resistance to bortezomib-induced apoptosis of RPMI8226 cells via activation of various intracellular pathways, including Jun N-terminal kinase, p38, p53, and Akt [76]. Similarly, our group has demonstrated the involvement of MM FBs-EVs in drug resistance. Specifically, MM cells co-cultured with MM FBs-EVs selectively overexpress miR-214-3p and miR-5100, which trigger MM cells proliferation and resistance to bortezomib-induced apoptosis via activation of intracellular pathways involved in cell apoptosis and proliferation, i.e., MAPK, AKT/mTOR, p53 (unpublished data).

## 5. EVs in MM Bone Resorption

Bone disease is a common feature of MM and it is associated with severe pain, pathologic fracture, spinal cord compression, vertebral collapse, and hypercalcemia [77]. It is characterized by the disruption of the physiological balance between osteoblasts (OBs) and osteoclasts (OCs), leading to a decrease in bone formation and an increase in bone resorption resulting in lytic lesions [77].

MM cells are important regulators of bone disease via cell-to-cell contact with BMSCs, which activate intracellular pathways involved in the regulation of OCs and OBs activities [78]. Recent studies demonstrated the involvement of MM-EVs in bone disease via the transfer of miRNAs, lncRNAs, and cytokines.

Raimondi et al. [79] demonstrated the ability of MM-EVs to trigger differentiation of OCs and promote bone resorption in both mice and humans. Specifically, MM-EVs induce the migration of OCs precursors (pOCs) by increasing the expression of CXCR4, and promoting their differentiation into multinuclear OCs capable to excavate bone lacunae. The uptake of EVs in pOCs induces the expression of markers of bone resorption activity, including tartrate-resistant acid phosphatase (TRAP), proteolytic enzyme cathepsin K (CTSK) and matrix metalloproteinases 9 (MMP-9). In addition, MM-EVs promote the survival of pOCs and inhibit their apoptosis [79].

Other studies documented the osteoclastogenic effect of EVs via direct and indirect mechanisms [80,81]. Proteomic analysis of MM1.s-EVs revealed that they directly contribute to bone resorption by containing several proteins involved in the unfolded protein response (UPR) pathway via the IRE1α/XBP1 axis. Selective inhibition of IRE1α partially counteracts EVs-induced OCs differentiation and bone resorption [80]. In addition, MM1.s-EVs contain the EGFR ligand amphiregulin (AREG), which directly activates the EGFR pathway in recipient pOCs cells, triggering OC differentiation. AREG-enriched EVs from MM cells activate OCs differentiation indirectly via MSCs. Indeed, MSCs uptake MM-EVs and release the pro-osteoclastogenic cytokine IL-8, which inhibits OBs differentiation by reducing the expression of OBs markers, i.e., alkaline phosphatase (ALP), osteocalcin (OCN), collagen type I alpha 1 (COL1A1) [81]. A recent study has shown that MM-EVs can contribute to bone resorption and lytic lesions by delivering the pro-inflammatory cytokine IL-32. IL-32 represents a poor prognostic factor that correlates with osteoclast activity and lytic lesions in MM patients and negatively with progression free survival [82].

RUNX2-AS1 is a natural antisense transcript derived from intron 7 of the RUNX2 gene, the major transcription factor associated with OB differentiation [83,84]. RUNX2-AS1 binds to RUNX2 pre-mRNA and affects the splicing of RUNX2, reducing its expression [83,84]. MM-EVs contain the lncRNA *RUNX2-AS1*, which decreases the expression of RUNX2 in MSCs as well as their osteogenic potential by regulating the expression of osteopontin (OPN) [85]. The osteolytic effect of MM-EVs was confirmed in vivo using the 5TGM1 mouse model: EVs significantly increase differentiation of OCs and improve their resorptive activity by reducing trabecular bone volume. Interestingly, analysis of EVs from the RPMI8226 MM cells revealed the presence of OPN (personal data not shown), which may be involved in bone resorption and support disease progression via angiogenesis [83].

Moreover, 5TGM1-derived EVs express the Wnt ligand DKK-1. DKK-1 downregulates Wnt signaling and negatively regulates OBs differentiation by decreasing the expression of master regulator genes for OBs differentiation, including RUNX2, Osterix, Col1A1 and ALP [78]. Liu et al. [70] demonstrated that MM-EVs inhibit the differentiation of BMSCs in OB by decreasing the mRNA levels of Ocn, Osterix, and Runx2. Co-culture of BMSCs with MM-EVs significantly reduced the number of OBs as well as their activity [70]. Moreover, EVs support bone disease by reprogramming the expression profile of OBs and OCs through miRs transfer. Raimondo et al. [86] demonstrated the overexpression of miR-129-5p in EVs from BM plasma of MM patients compared to SMM patients. Interestingly, co-cultures of MSCs with MM-EVs determine an increase in miR-129-5p, which inhibits the transcription factor Sp1, a positive modulator OB differentiation, and of its target gene Alp [86].

Collectively, these data suggest that the miRs cargo of MM-EVs may support bone disease during MM progression (Figure 2).

## 6. EVs in Immunosuppression

Immune dysregulation is a hallmark of MM and has been associated to disease progression from MGUS to symptomatic MM [87]. Immune dysfunction in MM patients includes quantitative, phenotypic, and functional abnormalities in dendritic cells (DCs), T cells, natural killer cells (NK), T regulatory cells (Treg), and myeloid-derived suppressor cells (MDSCs) [87,88,89,90,91]. MDSCs represent an immature myeloid cell population capable of repressing both innate and adaptive immunity [92]. MM patients exhibit increased levels of peripheral and BM MDSCs, suggesting that they have a role in immune escape of MM cells [93]. Uptake of MM- and BMSCs-EVs supports the expansion of murine MDSCs [94,95]. Both BMSCs- and MM-EVs activate Stat1 and Stat3 in target MDSCs and increase the levels of anti-apoptotic proteins Mcl-1 and Bcl-XL, demonstrating that EVs directly modulate the survival and apoptotic pathways of MDSCs [94,95]. Accordingly, inhibition of Stat1/3 pathways reduced MDSCs survival. In vivo studies confirmed the ability of BMSCs-EVs to maintain BM immunosuppression by inducing the expansion of MDSCs, which in turn suppress T cell proliferation and function through the release of nitric oxide (NO). Analysis of BM immune cells from naïve mice inoculated with 5T33MMvt EVs showed an increase of MDSCs, immature myeloid cells and eosinophils resembling the BM immune features of 5T33MM tumor bearing mice, implying that MM cells trigger BM immunosuppression via the release of EVs [94,95].

In addition, EVs contribute to BM immunosuppression by producing the immunosuppressive molecule adenosine (ADO). ADO is produced by adenosinergic ectoenzymes, i.e., CD39/CD73 or CD38/CD203a (PC-1)/CD73 from ATP or NAD^+^, respectively [96]. Morandi et al. [96] have shown that EVs from MM patients express higher surface levels of adenosinergic ectoenzymes, including CD38, CD39, CD73, and CD203a and, thus, resemble the phenotype of parental cells. In vitro analysis of EVs from MM, MGUS and SMM patients showed that MM-EVs efficiently metabolize ADO precursors, resulting in increased ADO production compared to MGUS and SMM. Interestingly, increased ADO levels as well as expression of CD38 on MM-EVs support resistance to the anti-CD38 monoclonal antibody treatment [97,98].

Thus, EVs from MM patients exert their immunomodulatory effect by maintaining the expansion of MDSCs and the production of the immunosuppressive molecule ADO (Figure 2).

## 7. EVs in Angiogenesis

The pleiotropic effect of EVs in MM also affects angiogenesis. Several studies documented the ability of BM-EVs to promote BM angiogenesis by transferring cytokines and miRs [94,99,100] (Figure 2).

MM-EVs from human RPMI8226 and murine 5T33MMvt cells contain pro-angiogenic proteins, i.e., angiogenin, HGF, MMP-9, VEGF, PDGF and SDF1-1α ([95] and personal data). They modulate cell signaling pathways, including Stat3, Jnk, and p53 in recipient BMSCs and STR10 endothelial cells (ECs), resulting in enhanced cell viability. Murine MM-EVs trigger angiogenesis in the chorioallantoic membrane (CAM) assay by promoting capillary and vessel formation [95].

The BM milieu of MM patients is characterized by an aberrant expression of HIF-1α and by the presence of hypoxic regions that further stimulate angiogenesis [101,102]. Umezu et al. [99] demonstrated the strong ability of hypoxia-resistant MM cells to induce angiogenesis by using an in vitro model that reproduced the hypoxic conditions of MM BM stroma. Hypoxia resistant MM cells release higher levels of EVs than normal MM cells and promote in vitro and in vivo angiogenesis through the delivery of miR-135b. In turn, miR-135b promotes the activation of hypoxia-induced angiogenic response by downregulating the factor inhibiting HIF-1 (FIH-1), a protein that binds to and inhibits the function of HIF-1α. Inhibition of FIH-1 leads to an increase in HIF-1α levels in target ECs and promotion of angiogenesis under hypoxic conditions [99]. Moreover, miR-let-7c embedded into MSCs-EVs promotes angiogenesis via M2 polarization of BM-resident and peripheral blood macrophages by inducing overexpression of CD206 and of miR-let-7c. M2 macrophages in turn trigger angiogenesis in vitro in a miR-let-7c–dependent manner [103].

Furthermore, Li et al. [100] have shown that EVs derived from MM cells, as well as circulating EVs, contain high levels of the small PIWI-interacting RNA 823 (piRNA-823), which correlates with disease progression, poor prognosis, and angiogenesis. Transfer of piRNA-823 into recipient EA.hy926 ECs prevents their apoptosis and increases ECs proliferation, invasion, angiogenesis, and tumor growth in vivo. Based on these results, it can be speculated that piRNA-823 may represent a therapeutic target to counteract angiogenesis and tumor growth.

## 8. Diagnostic Potential of EVs

Several studies have aimed to identify non-invasive biomarkers that are useful for disease diagnosis, prediction of therapeutic intervention, prognosis, and monitoring of disease progression.

The different EVs content among MM, MGUS, and HD patients, as well as their systemic circulation in all body fluids, including plasma, serum, and urine, make EVs suitable for non-invasive liquid biopsy.

Manier et al. [62] first suggested the prognostic significance of circulating EVs miR-let7-b and miR-18a in patients with MM [62]. Moreover, comparison of serum-derived miRs from MM, SMM and HD patients identified miR-140-3p, miR-20a-5p, miR-185-5p, and miR-4741 as oncogenic miRs highly correlated with malignant transformation, suggesting that they may serve as circulating biomarker of disease progression [63]. Similarly, Sedlarikova et al. [104] identified lncRNA *PRINS* as differentially expressed in MGUS, MM patients and HD, indicating the diagnostic potential of lnc*PRINS* [104].

Circulating EVs might also provide information on the efficacy of therapeutic strategies and could be used as biomarkers for primary and acquired drug resistance. Zhang et al. [105] found a correlation between downregulation of EVs miRs (miR-16-5p, miR-15a-5p and miR-20a-5p, miR-17-5p) and bortezomib resistance, indicating the potential use of EVs as circulating biomarkers for therapy management of MM patients [105]. Moreover, MSCs-EVs contain high levels of PSMA3 mRNA and lncRNA *PSMA3-AS1,* which mediate resistance to proteasome inhibitors and correlate to poor prognosis [106].

Due to the great diagnostic potential of EVs, highly sensitive methods, such as flow cytometry, digital PCR, microscopic imaging, and Raman spectroscopy, have been used to identify of EVs and define their cargo [107].

Marchisio et al. [108] described a method based on polychromatic flow cytometry to identify circulating EVs derived from different cells, i.e., ECs, leukocytes and platelets [108]. Applying this approach to MM could help to define the origin of circulating EVs based on the expression of specific surface antigens, i.e., CD38 for MM-EVs [98], CD31 for ECs-EVs [108]. Therefore, an increase in CD38^+^ and CD31^+^ EVs in the peripheral blood could predict the progression of MGUS to MM as well as the therapeutic response; thus, avoiding the BM biopsies of patients.

In addition, Raman spectroscopy and surface-enhanced Raman scattering were used to distinguish EVs from MGUS, asymptomatic and symptomatic MM based on their spectra. Multivariate analysis of these spectra should improve stratification of MM patients and their follow-up and risk of progression [109]. Recently, Laurenzana et al. [110] presented a new method for isolating EVs from peripheral blood in a single centrifugation step. They applied this method to characterize EVs from HD and MM patients by analyzing the size, concentration, and genetic content of EVs. The authors demonstrated increased levels of CD38^+^CD138^+^ EVs in the sera of MM patients. Interestingly, the number of CD38^+^CD138^+^ EVs correlates with plasmacytosis and disease stage [110].

Overall, these studies highlight the promising role of EVs as novel biomarkers for distinguishing clinical disease phase, monitoring MM progression and patient outcome, and predicting the efficacy of therapeutic strategies.

## 9. Therapeutic Perspective

Since EVs are known to play an important role in MM progression, several studies have focused on inhibiting EVs-mediated crosstalk by blocking the release and/or uptake of EVs to prevent their tumor-supportive activity [111] (Figure 3A).

Thompson et al. [112] demonstrated that heparanase induces release of EVs by tumor cells and affects their protein cargo by increasing the levels of syndecan-1, VEGF, and HGF [112]. Inhibition of heparanase activity through SST0001 suppresses MM cell growth and angiogenesis [113] (Figure 3A). The sphingolipid C6 ceramide affects MM cell proliferation, apoptosis, and EV release, and increases the levels of tumor-suppressive miRs, including miR-202, miR-16, miR-29b, and miR-15a embedded in MM-EVs [114].

GW4869, a neutral sphingomyelinase that prevents EVs budding from the plasma membrane [115], is cytotoxic for several MM cell lines and primary MM cells by binding phosphatidylserine expressed on their surface. Moreover, GW4869 is able to retard the growth of MM cells expressing phosphatidylserine in a mouse xenograft model [115]. Treatment of 5TGM1 mice with GW4869 reduces osteolysis by increasing OB activity and synergizes with bortezomib, leading to a reduction in tumor growth and microvessel density [116] (Figure 3A).

Inhibition of MM-EVs endocytosis by BMSCs by various endocytosis inhibitors (i.e., heparin, dynasore and omeprazole), prevents activation of intracellular pathways, i.e., STAT1, STAT3, and ERK1/2, which are associated with cell survival and proliferation [117]. Moreover, Tu et al. [118] have shown that EVs confer resistance to bortezomib. Blockade of endocytosis by chemical inhibitors reduces the internalization of BMSCs-EVs and sensitizes MM cells to bortezomib treatment in vitro and in vivo [118]. Other studies have explored the possibility of enhancing the EVs immune activity [119,120]. EVs derived from membrane-bound HSP70- or TNFα-engineered MM cells stimulate the anti-MM immune response in wild-type BALB/c mice by inducing DCs maturation [119,120]. Recently, Wang et al. [121] demonstrated that apoptotic EVs (apoEVs) derived from MSCs induce MM cells apoptosis and inhibit tumor growth by Fas pathway activation, suggesting a potential use of apoEVs as an anti-MM therapy (Figure 3B) [121].

Finally, since EVs transfer proteins and nucleic acids to neighboring and distant cells, they can be used as new therapeutic tools to deliver drugs, miRs, or lncRNAs to tumor cells [122]. To date, there are no anti-MM therapies based on EVs as a therapeutic agent. However, ongoing studies are investigating the possibility of encapsulating chemotherapeutic agents, e.g., doxorubicin, to ensure more specific drug delivery to tumor cells without side effects [123]. This strategy could be used for various anti-MM drugs, including monoclonal antibodies (e.g., anti-CD38, anti-SLAMF7 mAbs) to improve their efficacy [124]. Synthetic nanoparticles formulation has been developed for miRs delivery. The miR-34 based-therapy MRX34 was the first miR therapy in its class to be approved for the treatment of solid and hematological cancers, including MM. MRX34 delivers a miR-34 mimic encapsulated in a liposomal nanoparticle that causes miR-34 overexpression. Although the clinical trial was discontinued due to immune-mediated adverse effects, it provides evidence for the concept of miR delivery by EVs for cancer therapy [125].

## 10. Concluding Remarks

EVs are important mediators of intercellular communication by facilitating bidirectional crosstalk between cancer cells and tumor microenvironments in several diseases, including MM. The aberrant cargo packaged in the EVs isolated from the MM BM milieu is able to directly reprogram recipient cells and promote bone disease, angiogenesis, immunosuppression, and drug resistance, ultimately leading to tumor progression (Table 1).

The different EV cargo observed in pre-malignant and malignant stages, as well as the presence of EVs in almost all body fluids, makes them ideal candidates for non-invasive liquid biopsy to differentiate MGUS from overt MM, monitor MM progression, predict patient outcome, and predict the efficacy of therapeutic strategies.

Although the pathogenic role of EVs in MM is now widely documented, their application as a novel drug delivery tool is still in its infancy, and further investigation is needed to identify therapeutic strategies based on EVs.

## Figures and Tables

**Figure 1 cells-10-03185-f001:**
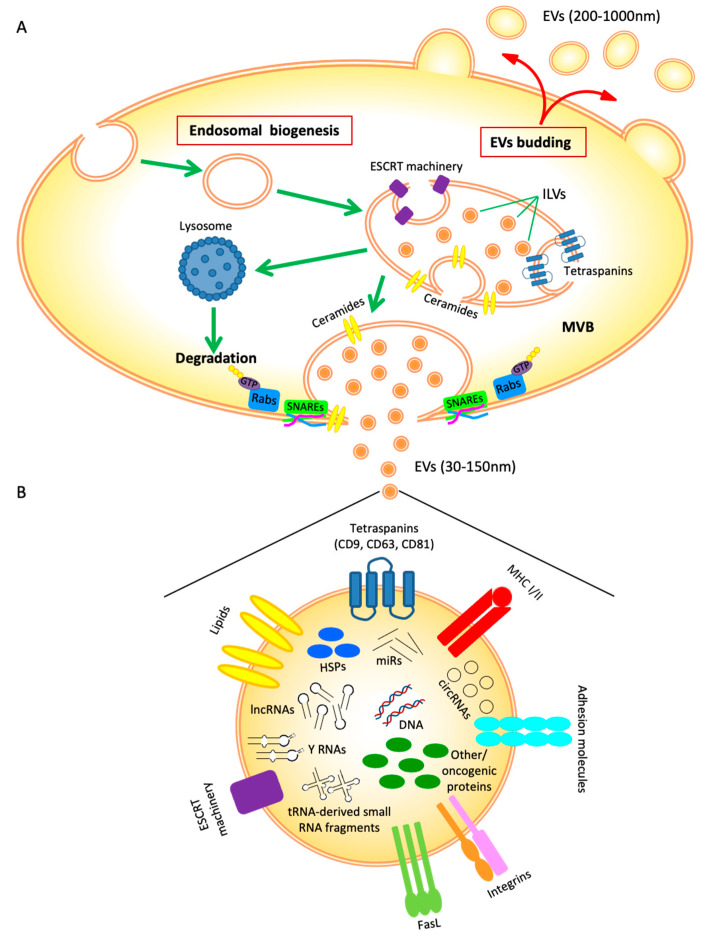
Schematic representation of EV biogenesis (**A**) and cargo (**B**). For more details, see the main text.

**Figure 2 cells-10-03185-f002:**
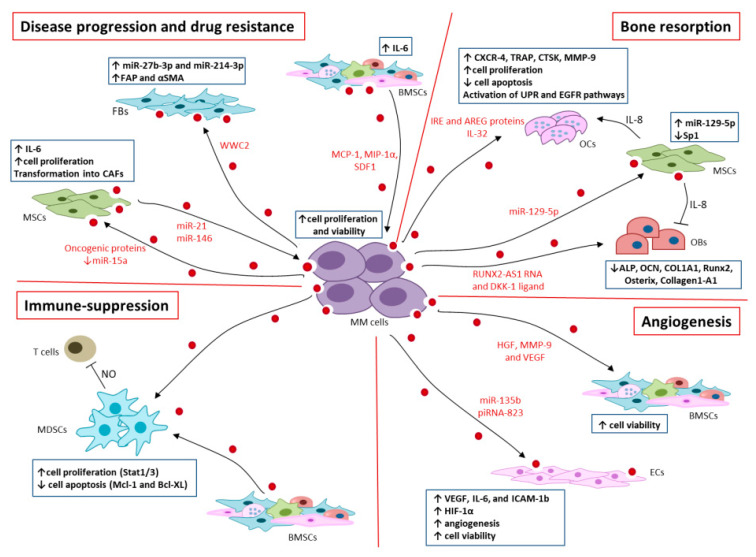
Schematic representation EVs-mediated interaction in BM microenvironment. MM cells release EVs that contain oncogenic proteins, cytokines, miRs, and other ncRNAs that support: (i) disease progression and drug resistance supporting fibroblast (FBs) activation, conversion of mesenchymal stromal cells (MSCs) into cancer-associated fibroblasts (CAFs) and IL-6 release by bone marrow stromal cells (BMSCs); (ii) bone resorption by increasing proliferation and function of osteoclasts (OCs) and preventing their apoptosis, and by inhibiting osteoblast differentiation; (iii) angiogenesis through the release of pro-angiogenic cytokines and chemokines, which promote the viability of endothelial cells (ECs) and BMSCs and the release of pro-angiogenic factors; and (iv) immunosuppression, which promotes the expansion of myeloid derived suppressor cells (MDSCs) and the production of the immunosuppressive adenosine (ADO). Finally, EVs from BMSCs, in turn, promote the proliferation and viability of MM cells, thus supporting disease progression and drug resistance.

**Figure 3 cells-10-03185-f003:**
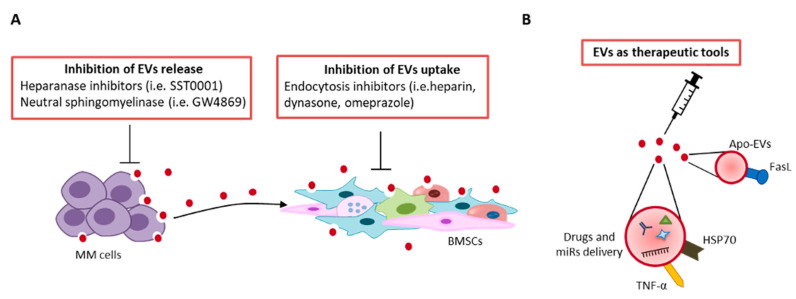
Schematic representation of EV therapeutic perspectives: (**A**) inhibition of EVs release and uptake, (**B**) EVs as therapeutic tools. For more details see the main text.

**Table 1 cells-10-03185-t001:** Biological and clinical relevance of role of EVs in MM.

Cargo Molecule	EVs Source	Target Cells	Biological Effect in Target Cells	Studies Model	Clinical Relevance	Clinical Application	Ref
WWC2	MM cells	FBs	Hippo pathway activation; miR-27b-3p and miR214-3p	In vitro	Transition MGUS to MM	Prognostic *	[68]
IL-6, MCP-1, fibronectin, miR-15a	MSCs	MM cells	Tumor growth	In vitro and in vivo	Tumor progression	Prognostic *	[74]
MCP-1, MIP-1α, SDF-1	BMSCs	MM cells	Activation of JNK, p38, p53, AKT pathways	In vitro	Tumor progression	Prognostic *	[76]
RUNX2-AS1	MM cells	MSCs	Down-regulation of RUNX2 expression and of osteogenic potential	In vitro and in vivo	Tumor-associated bone loss	Prognostic *	[85]
Not specified	BMSCs, MM cells	MSDCs	Increased STATs-mediated viability	In vitro and in vivo	Immunosuppression	Prognostic *	[94,95]
miR-129-5p	MM cells, BM plasma	MSCs	Down-regulation of Sp1 and reduced osteogenic differentiation	In vitro and ex vivo	Bone lesions, transition SMM to MM	Prognostic	[86]
Angiogenin, HGF, MMP-9, VEGF	MM cells	ECs	Enhanced viability	In vitro and in vivo	Increased angiogenesis	Prognostic	[95]
IL32	MM cells	preOCs	Increased osteoclast activity	In vitro and ex vivo	Osteolytic bone disease, reduced PFS	Prognostic	[82]
miR-135b	Hypoxic MM cell lines	ECs	Downregulation FIH-1	In vitro and in vivo	Increased angiogenesis	Prognostic	[99]
let-7b and miR-18a	Plasma	Not specified	Not specified	Ex vivo	Negative correlation with PFS and OS	Prognostic	[62]
Adenosinergic ectoenzymes	BM plasma	Not specified	Increased ADO	Ex vivo	Immunosuppression, HD to MM transition	Diagnostic	[96]
miR-20a-5p, miR-103a-3p, miR-4505	Serum	Not specified	Not specified	Ex vivo	HD and MM transition transition	Diagnostic	[63]
Let-7c-5p, miR-185-5p, miR-4741	Serum	Not specified	Not specified	Ex vivo	SMM to MM transition	Diagnostic	[63]
lncRNA PRINS	Serum	Not specified	Not specified	Ex vivo	Correlation with biochemical parameters in MGUS and MM patients	Diagnostic	[104]
PSMA3 mRNA, PSMA3-AS1	MSCs	Not specified	Increased PSMA3 protein levels and increased proteasome activity	In vitro and ex vivo	Resistance to proteasome inhibitors, high levels correlating to OS and PFF	Prognostic, Response to therapy	[106]
miR-16-5p, miR-15a-5p, miR-20a-5p, miR-17-5p	Serum	Not specified	Not specified	Ex vivo	Resistance to bortezomib	Prognostic, Response to therapy	[105]
UPR proteins	MM cells	Macrophages	OCs terminal differentiation	In vitro	Bone resorption	Prognostic, Therapeutic target	[80]
piRNA-823	MM cells, serum	ECs	Increased survival, proliferation, and angiogenesis	In vitro and ex vivo	Increased angiogenesis, poor prognosis	Prognostic, Therapeutic target	[100]

Summary of the cited studies on the role of EVs in MM. The specific EVs cargo composition, source, EVs molecular and biological effects, and their clinical implications are listed. * Prognostic value of the specific cargo molecule wasn’t directly addressed in the indicated study. SMM: Smouldering Multiple Myeloma; MM: Multiple Myeloma; MGUS: Monoclonal gammopathy of undetermined significance; OCs: Osteoclasts; PFS: Patient Free Survival; MSCs: Mesenchymal Stromal Cells, BMSCs: Bone Marrow Stromal Cells; MDSCs: Myeloid-Derived Suppressor Cells; ADO: Adenosine; ECs: Endothelial Cells; OS: Overall Survival.

## Data Availability

Not applicable.

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
