# Peer review of "Role of Extracellular Vesicle-Based Cell-to-Cell Communication in Multiple Myeloma Progression"

_cells, 2021, doi:10.3390/cells10113185_

Round 1
Reviewer 1 Report
In this review, Saltarella and colleagues describe the role of extracellular vesicles in multiple myeloma, in particular in EVs derived from the bone marrow and how they influence the tumour microenvironment. Although the review is of an appealing length and contains a reasonable amount of information, it still needs substantive editing to be finally accepted for publication. The quality of the English language used in the manuscript needs improvements.
In particular, the EV section including biogenesis and cargo does not reflect the current state of research and appears mainly too superficial. Therefore, it needs to be revised and improved.
In the following, only some apparent issues will be pointed out:
- In the introduction to EVs, the authors do not always discuss the state of knowledge of current EV research. As an example, on page 1 from line 42, the difference between the various EVs subtypes based on the size can be mentioned. The statement in the review is not sustainable and should be corrected because the size (which is not always correct stated here) is not a valid criterion to distinguish between the different EVs.
- The statement that microvesicle budding is a rather random process is not correct
- In general, the chosen nomenclature is not in line with the consensus. Instead of referring to exosomes (EXOs), the authors should use the generic term EV. In addition, the abbreviation EXOs diminishes the reading flow in general.
- Even though the EXO biogenesis section is adequately described, Figure 1 should be revised as it seems too simplistic.
- Exosomes were already described initially from 1970 on and also called as such. Please also correct this.
- In the "EXO cargo" section, the authors, unfortunately, do not discuss the most recent state of research (e.g. protein corona and RNA in EVs which has already been (critically) described several times within this special issue. This should be updated. Figure 2, on the other hand, is well done.
Throughout the entire review, the authors do not always refer to the relevant original papers in their review, which would be desirable for a review. Therefore, please revise your references.
Typos, missing small words, etc. have to be corrected.
Author Response
POINT-BY-POINT ANSWERS TO REVIEWER 1 COMMENTS
Authors thank the Reviewer 1 for helpful criticism.
Reviewer’s comment 1-4:
- In the introduction to EVs, the authors do not always discuss the state of knowledge of current EV research. As an example, on page 1 from line 42, the difference between the various EVs subtypes based on the size can be mentioned. The statement in the review is not sustainable and should be corrected because the size (which is not always correct stated here) is not a valid criterion to distinguish between the different EVs.
- The statement that microvesicle budding is a rather random process is not correct
- In general, the chosen nomenclature is not in line with the consensus. Instead of referring to exosomes (EXOs), the authors should use the generic term EV. In addition, the abbreviation EXOs diminishes the reading flow in general.
- Even though the EXO biogenesis section is adequately described, Figure 1 should be revised as it seems too simplistic.
Reply: According to Reviewer’s comments, introduction has been revised. In particular, we have described the first classification and nomenclature of EVs based on their size and biogenesis:
“Based on their size, EVs were first classified into apoptotic/necrotic bodies (size 1000-5000nm), microvesicles (200-1000nm) and exosomes (30-150nm) [3]. Apoptotic/necrotic bodies originate from apoptotic and necrotic cells that break up into multiple vesicles, microvesicles are pinched off from the plasma membrane, while exosomes are formed by an active, energy-dependent and organized process [4].”
Next, based on the MISEV18 guidelines (Thery et al., Nat Rev Immul. 2018), we added a sentence with a new definition of EVs:
“Currently, the MISEV18 guidelines recommend the use of the term “EVs” for a heterogeneous family of vesicles (both small and medium/large vesicles) delimited by a lipid bilayer that are released from all eukaryotic cells in various body fluids (plasma, urine, saliva and cerebrospinal fluid)[5].” (lines 45-49)
We also used the generic term EVs to describe exosomes (EXOs) and microvesicles (MVs) throughout the entire review.
Finally, according to Reviewer’s suggestion, we have revised the first sentence on EVs biogenesis, which now states:
“Two main pathways of biogenesis have been described for the generation of EVs: i) direct budding of the plasma membrane with consequent release of EVs, classically referred to as microvesicles [4,12], and ii) endosomal biogenesis, an active and complex mechanism that ultimately leads to the production of EVs, classically referred to as exosomes (Figure 1A) [13].” (lines 70-74)
We have also modified Figure1. The new Figure1 includes 2 panels that describe the biogenesis (panel A) and cargo (panel B) of EVs.
Reviewer’s comment 5: Exosomes were already described initially from 1970 on and also called as such. Please also correct this.
Reply: We thank the Reviewer for this comment. We realized that the initial sentence was misleading, so we corrected the sentence as:
“Although exosomes were initially regarded as cellular waste with no biological activities [6], it is now widely accepted that EVs play an essential role in intercellular communication via their cargo, capable of reprogramming near and distal recipient cells [3,7].” (lines 50-53)
In this sentence, we want to highlight the evolving concept of EVs. Indeed, Johnstone et al (Johnstone et al. J Biol Chem. 1987) suggested that EVs secretion was a mechanism to remove cell components no longer needed to reticulocytes during maturation. By contrast, nowadays EVs play an important role in the cell-to-cell communication reprogramming recipient cell behavior.
Reviewer’s comment 6: In the "EXO cargo" section, the authors, unfortunately, do not discuss the most recent state of research (e.g. protein corona and RNA in EVs, which has already been (critically) described several times within this special issue. This should be updated. Figure 2, on the other hand, is well done.
Reply: According to Reviewer’s comment, the cargo section has been revised by discussing and including recent references, such as the new references 47-53, which describe the presence of protein corona around EVs:
“Recent studies have shown that EVs can be ‘decorated’ with additional proteins besides the canonical exofacial molecules, collectively referred to as ‘corona’ [46-52]. Buzas et al. [46] showed that EVs associate with extracellular matrix proteins, complement, immunoglobulins, coagulation factors, lipoproteins, nucleic acids and thiol‐reactive antioxidants [46]. Furthermore, Toth et al. [51] confirmed the interaction between plasma protein and EVs and identified a number of proteins (ApoA1, ApoB, ApoC3, ApoE, complement factors 3 and 4B, fibrinogen a-chain, immunoglobulin heavy constant g2 and g4 chains) that form a ‘corona’ around EVs in blood plasma [51].” (lines 133-140).
We did not discuss in detail the RNA content in EVs due to the high number of published studies. Indeed, our purpose was to give a general overview, since the RNA cargo of EVs has already been substantially and critically described. For this reason, we have only mentioned the diverse RNA contents of EVs (microRNAs –miRs-, long non-coding RNAs –lncRNAs-, circular RNAs –circRNAs-, tRNA-derived small RNA fragments and Y RNAs, for recent and detailed review see [57]).” (lines 144-145) and we suggested to refer to the review of Veziroglu et al. (Veziroglu et al., Front Genet. 2020) for a recent and detailed discussion.
Reviewer’s comment 7: Throughout the entire review, the authors do not always refer to the relevant original papers in their review, which would be desirable for a review. Therefore, please revise your references.
Reply: According to Reviewer’s comment, we have revised reference section.
Reviewer’s comment 8: Typos, missing small words, etc. have to be corrected.
Reply: The review has been revised by an English proof-reader.
Reviewer 2 Report
In their review, Saltarella and others discuss the role of EVs and EV-targeted therapeutic interventions in multiple myeloma, a neoplasm of bone marrow cells. They provide a thorough primer on EV biology then cover the role of EVs in disease progression and the approaches that target blunting EV signaling among tumor cells. Finally they discuss ways in which EVs themselves can be used as drug delivery platforms to target tumor cells.
On the whole this is a thoughtfully written review and I commend the authors on the breadth of the material coverage.
As English is likely not the first language of the authors, there are some errors in word choice and grammar (not attributed to carelessness). Those include but are not limited to:
Line 47: "Energy-needing" should be, instead, be "energy-dependent"
Line 64: The sentence should read "...in the reprogramming OF near and distant"
Line 80: The sentence should read: "...consists OF" not "...consists IN"
Line 200: "In details,..." is awkward wording, perhaps "Specifically,..." would be more appropriate.
The only other comment I have is that it would be helpful if the authors provided a summary figure at the end of the paper showcasing the roles of EVs in disease progression and therapy (including diagnostic and therapeutic applications).
Author Response
POINT-BY-POINT ANSWERS TO REVIEWER 2 COMMENTS
Authors thank the Reviewer 2 for helpful criticism and are glad for positive comments.
Reviewer’s comment 1: As English is likely not the first language of the authors, there are some errors in word choice and grammar (not attributed to carelessness). Those include but are not limited to:
Line 47: "Energy-needing" should be, instead, be "energy-dependent"
Line 64: The sentence should read "...in the reprogramming OF near and distant"
Line 80: The sentence should read: "...consists OF" not "...consists IN"
Line 200: "In details,..." is awkward wording, perhaps "Specifically,..." would be more appropriate.
Reply: We thank the Reviewer for this suggestion. Accordingly, these sentences have been modified and the review has been revised by an English proof-reader.
Reviewer’s comment 2: The only other comment I have is that it would be helpful if the authors provided a summary figure at the end of the paper showcasing the roles of EVs in disease progression and therapy (including diagnostic and therapeutic applications).
Reply: We agree with the Reviewer. In line with its advice, we added Table 1 that summarizes the biological and clinical relevance of EVs in MM (including diagnostic and therapeutic applications), and Figure 3 that depicts the main therapeutic applications of EVs.
Reviewer 3 Report
The authors review the role of EVs derived from MM cells/BM cells in modifying the microenvironment through the induction of bone resorption, immunosuppression and angiogenesis. The manuscript is well structured and well presented; however, I am not sure if it adds to the literature, since there are numerous similar reviews on this topic in the current literature ("extracellular vesicles" OR exosomes OR microvescicles) AND myeloma - Search Results - PubMed (nih.gov). The authors should better describe the rationale of their review despite all the available reviews in the literature. Are there any new data that have not been previously discussed? Light grammar and syntax editing is needed. Can the authors speculate on the potential integration of EVs as biomarkers for response and/or tools for decision making regarding the management of patients with myeloma? Can the authors provide a Table presenting the differences in EVs expression in MGUS as compared with MM?
Author Response
POINT-BY-POINT ANSWERS TO REVIEWER 3 COMMENTS
Authors thank the Reviewer 3 for helpful criticism and are glad for positive comments.
Reviewer’s comment 1: The authors review the role of EVs derived from MM cells/BM cells in modifying the microenvironment through the induction of bone resorption, immunosuppression, and angiogenesis. The manuscript is well structured and well presented; however, I am not sure if it adds to the literature, since there are numerous similar reviews on this topic in the current literature ("extracellular vesicles" OR exosomes OR microvesicles) AND myeloma - Search Results - PubMed (nih.gov). The authors should better describe the rationale of their review despite all the available reviews in the literature. Are there any new data that have not been previously discussed?
Reply: We thank the Reviewer for this comment. Accordingly, we described new data published in recent months (references 71, 75, 103, 121) and we emphasized the promising therapeutic perspective based on EVs in MM and their diagnostic potential (see reply to Reviewer’s comment 3). The most recent updates are described as follow:
“Finally, EVs from both plasma and MM cells of patients promote cell proliferation, migration, and adhesion of recipient BMSCs by transferring multiple oncogenic factors highlighting the role of MM-EVs in modifying the BM niche [71].” (lines 185-188)
“Recently, proteomic analysis of MSCs-EVs cargo revealed that MM MSCs-EVs heterogeneously express higher levels of ribosomal proteins than their HD counterpart. EVs enriched with ribosomal proteins promote the viability and proliferation of MM cells [72,75]” (lines 198-201).
“Moreover, miR-let-7c embedded into MSCs-EVs promotes angiogenesis via M2 polarization of BM-resident and peripheral blood macrophages by inducing overexpression of CD206 and of miR-let-7c. M2 macrophages in turn trigger angiogenesis in vitro in a miR-let-7c–dependent manner [103]. ” (lines 322-326)
“Recently, Wang et al. [121] demonstrated that apoptotic EVs (apoEVs) derived from MSCs induce MM cells apoptosis and inhibit tumor growth by Fas pathway activation, suggesting a potential use of apoEVs as an anti-MM therapy (Figure 3B) [121].” (lines 409-411)
Reviewer’s comment 2: Light grammar and syntax editing is needed.
Reply: According to Reviewer’s comment, the review has been revised by an English proof-reader.
Reviewer’s comment 3: Can the authors speculate on the potential integration of EVs as biomarkers for response and/or tools for decision making regarding the management of patients with myeloma?
Reply: We thank the Reviewer for this advice. Accordingly, a new “Diagnostic potential of EVs” section has been added (lines 335-377). In this section, we describe the diagnostic potential of EVs as novel biomarker able to discriminate clinical disease phase, patient’s outcome and to foresee the efficacy of therapeutic strategies. Moreover, we discuss highly sensitive methods such as Raman spectroscopy, surface-enhanced Raman scattering, polychromatic flow cytometry to determine the identity of EVs and define their cargo. The new section states:
“Diagnostic potential of EVs
Nowadays, several studies aim to identify non-invasive biomarkers useful for disease diagnosis, prediction of therapeutic intervention, prognosis and monitoring of disease progression.
The different EVs content between MM, MGUS and HD patients as well as their systemic circulation in all body fluids including plasma, serum and urine make EVs suitable for non-invasive liquid biopsy.
Manier et al. [62] first suggested the prognostic significance of circulating EVs miR-let7-b and miR-18a in patients with MM [62]. Moreover, comparison of serum-derived miRs from MM, SMM and HD patients identified miR-140-3p, miR-20a-5p, miR-185-5p, and miR-4741 as oncogenic miRs highly correlated with malignant transformation, suggesting that they may serve as circulating biomarker of disease progression [63]. Similarly, Sedlarikova et al. [104] identified lncRNA PRINS as differentially expressed in MGUS, MM patients and HD, indicating the diagnostic potential of lncPRINS [104].
Circulating EVs may also provide information on the efficacy of therapeutic strategies and can be used as biomarkers for primary and acquired drug resistance. Zhang et al. [105] found a correlation between downregulation of EVs miRs (miR-16-5p, miR-15a-5p and miR-20a-5p, miR-17-5p) and bortezomib resistance, indicating the potential use of EVs as circulating biomarkers for therapy management of MM patients [105]. Moreover, MSCs-EVs contain high levels of PSMA3 mRNA and lncRNA PSMA3-AS1, which mediate resistance to proteasome inhibitors and correlate to poor prognosis [106].
Due to the great diagnostic potential of EVs, highly sensitive methods such as flow cytometry, digital PCR, microscopic imaging, and Raman spectroscopy have been used to determine identity of EVs and define their cargo [107].
Marchisio et al. [108] described a method based on polychromatic flow cytometry to identify circulating EVs derived from different cells, i.e. ECs, leukocytes, platelets [108]. Applying this approach to MM could help to define the origin of circulating EVs based on the expression of specific surface antigens, i.e. CD38 for MM-EVs [98], CD31 for ECs-EVs [108]. Therefore, an increase in CD38+ and CD31+ EVs in the peripheral blood could predict the progression of MGUS to MM as well as the therapeutic response, thus avoiding the BM biopsies of patients.
In addition, Raman spectroscopy and surface-enhanced Raman scattering were used to distinguish EVs from MGUS, asymptomatic and symptomatic MM based on their spectra. Multivariate analysis of these spectra should improve stratification of MM patients and their follow-up and risk of progression [109]. Recently, Laurenzana et al. [110] presented a new method for isolating EVs from peripheral blood in a single centrifugation step. They applied this method to characterize EVs from HD and MM patients by analyzing the size, concentration, and genetic content of EVs. The authors demonstrated an increased levels of CD38+CD138+ EVs in the sera of MM patients. Interestingly, the number of CD38+CD138+ EVs correlates with plasmocytosis and disease stage [110].
Overall, these studies highlight the promising role of EVs as novel biomarkers for distinguishing clinical disease phase, monitoring MM progression and patients outcome and predicting the efficacy of therapeutic strategies.”
Reviewer’s comment 4: Can the authors provide a Table presenting the differences in EVs expression in MGUS as compared with MM?
Reply: We agree with the Reviewer’s suggestion. In line with this advice, we added a new Table 1 that summarizes the biological differences between MGUS and MM EVs and the clinical relevance of EVs in MM.
Round 2
Reviewer 3 Report
The authors have addressed the reviewers' comments